# Sar1 Affects the Localization of Perilipin 2 to Lipid Droplets

**DOI:** 10.3390/ijms23126366

**Published:** 2022-06-07

**Authors:** Tomohiko Makiyama, Takashi Obama, Yuichi Watanabe, Hiroyuki Itabe

**Affiliations:** Division of Biological Chemistry, Department of Pharmaceutical Sciences, School of Pharmacy, Showa University, 1-5-8 Hatanodai, Shinagawa, Tokyo 142-8555, Japan; obama@pharm.showa-u.ac.jp (T.O.); yu.watanabe@pharm.showa-u.ac.jp (Y.W.); h-itabe@pharm.showa-u.ac.jp (H.I.)

**Keywords:** lipid droplet, Sar1, COPⅡ, perilipin 2, triacylglycerol

## Abstract

Lipid droplets (LDs) are intracellular organelles that are ubiquitous in many types of cells. The LD core consists of triacylglycerols (TGs) surrounded by a phospholipid monolayer and surface proteins such as perilipin 2 (PLIN2). Although TGs accumulate in the phospholipid bilayer of the endoplasmic reticulum (ER) and subsequently nascent LDs buds from ER, the mechanism by which LD proteins are transported to LD particles is not fully understood. Sar1 is a GTPase known as a regulator of coat protein complex Ⅱ (COPⅡ) vesicle budding, and its role in LD formation was investigated in this study. HuH7 human hepatoma cells were infected with adenoviral particles containing genes coding GFP fused with wild-type Sar1 (Sar1 WT) or a GTPase mutant form (Sar1 H79G). When HuH7 cells were treated with oleic acid, Sar1 WT formed a ring-like structure around the LDs. The transient expression of Sar1 did not significantly alter the levels of TG and PLIN2 in the cells. However, the localization of PLIN2 to the LDs decreased in the cells expressing Sar1 H79G. Furthermore, the effects of Sar1 on PLIN2 localization to the LDs were verified by the suppression of endogenous Sar1 using the short hairpin RNA technique. In conclusion, it was found that Sar1 has some roles in the intracellular distribution of PLIN2 to LDs in liver cells.

## 1. Introduction

Lipid droplets (LDs) are unique intracellular organelles that store triacylglycerols (TGs) or cholesterol ester [1,2]. LDs are ubiquitous in many types of cells and organs [3], but the proteins and lipids that form them often vary based on cell type [4]. In hepatoma cells, an LD consists mainly of TGs surrounded by a phospholipid monolayer and proteins associated with the LD surface, including perilipin 2 (PLIN2) [5,6]. TGs are synthesized in the endoplasmic reticulum (ER) from diacylglycerol (DG) and acyl-CoA by DG acyltransferase (DGAT) [7,8,9]. TGs’ accumulation between the two leaflets of the ER membrane bilayer generates a lens-like structure that grows to form a small extrusion and then buds off into the cytosol as a nascent LD [10,11]. Although an LD is derived from the ER, its phospholipid composition is quite different from that of the ER [5]. Proteins associated with LDs are divided into two classes [12,13]. Class I LD proteins, such as caveolin-1 and DGAT2, are transported to LDs from the ER, while Class Ⅱ LD proteins, such as phosphocholine cytidylyltransferase α (CCTα) and Cide-A, can be transferred to LDs from the cytosol [13,14]. However, the mechanism by which these LD proteins are transported to LDs during LD formation has not been fully understood.

LD protein profiles in many types of cells have been studied [15,16]. Perilipin (PLIN) family proteins are well studied as LD-associated proteins that possess conserved domains. Among them, PLIN2 is expressed ubiquitously in many types of cells and is a major LD protein in liver. PLIN2 binds tightly to LD although it does not contain typical transmembrane sequences. PLIN3, another member of PLIN family proteins widely expressed in many types of cells, distributes both in cytosol and LD. Lipid metabolizing enzymes are also typical members found in LD fractions, such certain types of as acyl-CoA synthase and hydroxysterosis dehydrogenase. Proteins related to vesicle trafficking, such as Rab proteins, or SNARE proteins, are frequently detected in isolated LD-fractions. In addition, Sar1b, a regulator coat protein complex Ⅱ (COPⅡ) vesicle traficking, was shown to be present in LD fractions in steatotic liver [17].

Newly synthesized membrane proteins and lipids are transported to the Golgi apparatus from the ER via COPⅡ-coated vesicles [18,19]. COPⅡ vesicle budding is facilitated by Sar1, an Arf family small GTPase, which is activated by Sec12, a guanine nucleotide exchange factor (GEF) [20]. Activated Sar1 binds to the ER membrane and recruits the Sec23/24 complex [21]. There are two isoforms of Sar1, Sar1a and Sar1b, that are similar enough that each can compensate for a deficiency in the other during COPⅡ vesicle budding [22,23]. Although the two isoforms have high sequence homology, there are some functional differences. For example, Sar1a exchanges GDP-GTP faster than Sar1b, and the affinity of Sar1b to Sec23 is higher than that of Sar1a [22,24]. *SAR1B* genetic defects cause chylomicron retention diseases, despite the presence of Sar1a, and they are characterized by a remarkable reduction of plasma TG levels and appearance of intracellular vesicles containing neutral lipids in enterocytes [24,25]. We believe that Sar1 could play a role in LD formation, because both COPⅡ vesicles and LD are generated from the ER membrane; however, the effect of Sar1 on LD remains unknown.

In this study, we investigated the effects of Sar1 on LDs using a human hepatoma cell line HuH7. Our data show that Sar1 is involved in PLIN2 localization to the LD, and that simultaneous abrogation of both isoforms of Sar1 disturbed the intracellular distribution of PLIN2.

## 2. Results

### 2.1. Peripheral Localization of GFP-Sar1b WT on Lipid Droplet in Oleic Acid-Treated Cells

There are two isoforms of Sar1: Sar1a and Sar1b. Because Sar1b protein level is higher than Sar1a level in the liver [22], we first investigated the role of Sar1b in LD formation in HuH7 cells. Previously, we showed that HuH7 cells accumulated PLIN2-coated LD upon incubation with oleic acid (OA), and the localization of PLIN2 to LD was demonstrated [26]. To examine the intracellular localization of Sar1b in OA-treated cells, we prepared adenovirus particles containing genes coding either GFP (control), GFP fused with wild-type Sar1b (GFP-Sar1b WT), or GFP fused with a mutant form of Sar1b (GFP-Sar1b H79G). The H79G mutant is used as a GTPase-deficient Sar1 mutant by a dominant negative effect. This type of mutant is a constitutive active form; however, it interferes with the conversion of GTP- and GDP-forms to bring about the inhibition of Sar1 function in cells. [27,28]. In cells expressing this type of Sar1 mutant, COPⅡ vesicle fails to detach from the ER; thus, the H79G mutant is considered a potent inhibitor of ER to Golgi transport. In cells expressing GFP, GFP was distributed in the whole-cell space, except in the LD area (Figure 1A, panel OA-a). GFP-Sar1b WT protein was also found throughout the cytosol; however, part of it accumulated at the periphery of the LD particles when the cells were treated with OA (Figure 1A panel OA-b). The distribution of GFP-Sar1b H79G showed a dot-like pattern and was not associated with LD particles after OA treatment (Figure 1A panel OA-c). The number and total area of LDs in cells expressing GFP, GFP-Sar1b WT, and GFP-Sar1b H79G were not significantly different (Figure 1B,C).

### 2.2. Intracellular PLIN2 Protein Levels Decrease in Sar1b H79G Cells

We investigated the intracellular levels of TG and the PLIN2 protein, the major constituents of LDs in the liver, using the cells transiently expressing either GFP, GFP-Sar1b WT, or GFP-Sar1b H79G. Intracellular TG levels were not changed by the expression of Sar1b or its mutant form (Figure 2A). However, PLIN2 expression decreased slightly but significantly in cells expressing GFP-Sar1b H79G, compared to that in cells expressing GFP or GFP-Sar1b WT (Figure 2B,D). Next, PLIN3, another member of the LD-associating PAT family proteins [29,30], was examined in these cells by western blotting. In contrast to PLIN2, PLIN3 and KDEL (an ER stress marker) expression levels did not change after the introduction of either of the Sar1b proteins in HuH7 cells (Figure 2B,C). Several proteins including PLIN2, PLIN3, and GFP-Sar1 fusion proteins were not recovered in insoluble fraction but in soluble fraction after centrifugation of cell lysate; thus, we assume that protein aggregates were not generated in the cells after overexpression of Sar1 proteins and OA treatment (Appendix A). Figure 2E shows that intracellular *PLIN2* mRNA levels were not significantly changed by the expression of WT or H79G Sar1b proteins either with or without OA treatment. Taken together, these results indicate that the amount of PLIN2 decreased in cells expressing the mutant form of Sar1b (H79G) without a change in its transcriptional level. One possibility is that PLIN2 may be degraded intracellularly, because we and others have previously shown that cellular PLIN2 levels are regulated not at the transcriptional level, but by degradation via the ubiquitin-proteasome pathway in cells under certain conditions [31,32].

### 2.3. Sar1b H79G Mutant Affects PLIN2 Localization to LDs

Next, we investigated the intracellular localization of PLIN2. Lysates of cells treated with OA were fractionated using sucrose density gradient centrifugation in which the post-nuclear supernatant of the cell lysate adjusted to 26% sucrose was placed in the middle of 2–51% discontinuous gradient. The sample was fractionated from the top after the centrifugation, and each fraction was analyzed by western blotting (Figure 3A). PLIN2 was mainly found in the top fractions (1 and 2), and we defined them as the ”fraction”. PLIN2 was also found in the middle fractions (5 and 6), and we defined these fractions as the “intermediate LD fraction”. The distribution of PLIN2 in HuH7 cells is consistent with our previous observation [26]. Other proteins, such as calnexin (an ER marker), LAMP1 (a lysosome marker), GM130 (a cis-Golgi marker), and Bcl-2 (a mitochondria marker) were found in the bottom fractions (12, 13, and 14). The endosome marker, EEA1, and GFP were recovered in fractions 8–12, indicating that the LD fraction had not been contaminated by other major organelles (Figure 3B). In cells expressing GFP-Sar1b WT, PLIN2 was mainly found in the LD fraction and partly in the intermediate LD fraction (Figure 3C). However, the PLIN2 level in the LD fraction decreased, and the distribution pattern was broader in cells expressing GFP-Sar1b H79G than in those expressing GFP or GFP-Sar1b WT (Figure 3B–D). In the cells expressing GFP-Sar1b WT, PLIN2 intensity in the intermediate LD fraction was more detectable than that in GFP expressing cells, but not significantly (*p*-Value is 0.204). In contrast to the distribution pattern for PLIN2, the PLIN3 distribution patterns in cells expressing GFP, GFP-Sar1b WT, or GFP-Sar1b H79G were constant. Previously, Sec22b was reported as one of COPⅡ vesicular membrane proteins [33], and we showed that Sec22b was present in LDs in mouse Leydig cell line MLTC-1 cells [15]. Sec22b was found in the LD fraction in cells expressing GFP-Sar1b WT, but not in the GFP-Sar1b H79G cells (Figure 3C). TG levels in the LD fraction did not change significantly among cells expressing GFP, GFP-Sar1b WT, or GFP-Sar1b H79G (Figure 3E).

PLIN2 distribution in these cells was examined by immunofluorescent staining (Figure 4). The PLIN2 signal was faint in cells without OA (Con), and it was much stronger in cells treated with OA (OA). In GFP-expressing cells treated with OA, the PLIN2 signal was found to distribute in cytosolic space and there were some ring-like structures where PLIN2 colocalize with GFP-Sar1b WT (Figure 4B(b)). In OA-treated GFP-Sar1b H79G expressing cells, however, the PLIN2 signal was blurred compared with that in cells expressing GFP or GFP-Sar1b WT, and colocalization of GFP-Sar1b H79G with PLIN2 was not observed. Line profile analysis supported these observations (Appendix A). These results, together with the results of sucrose density gradient centrifugation (Figure 3), suggest that Sar1 has a role in PLIN2 localization to LDs in cells that is lost in the cells expressing the H79G mutant form.

### 2.4. Similar Contribution of Sar1a on PLIN2 Localization to LDs

In addition to Sar1b, Sar1a plays a role in recruiting COPⅡ components, and Sar1a and Sar1b can each compensate for a deficiency in the other [22,23,34]. Thus, the effect of Sar1a on PLIN2 localization in the LD was studied. Both GFP-Sar1a WT and GFP-Sar1a H79G were found throughout the cytosol, but not in the nucleus (Appendix A). GFP-Sar1a WT, but not GFP-Sar1a H79G, had a ring-like structure at the LD peripheral area in OA-treated cells, similar to that of Sar1b (Appendix A). The number and total area of LDs in cells expressing GFP, GFP-Sar1a WT, and GFP-Sar1a H79G were not different (Appendix A). Intracellular TG levels did not change in cells expressing the WT or H79G Sar1a (Appendix A), which was consistent with the immunofluorescence results. Unlike the case with Sar1b, PLIN2 and PLIN3 levels in cell lysates did not change significantly upon the introduction of the mutant form of Sar1a (H79G) (Appendix A). When cell lysates were fractionated using sucrose density gradient centrifugation, PLIN2 and PLIN3 were found in the top fractions. However, the PLIN2 level in the LD fraction decreased in the cells expressing GFP-Sar1a H79G, and they were broadly distributed (Figure 5A,B).It is noted that distribution pattern of PLIN3 in GFP-Sar1a H79G expressed cells is slightly different from that in GFP-Sar1a WT expressed cells, where substantial amount of PLIN3 appeared in fractions 1-5 (Figure 5A). TG levels in the LD fraction were the same in cells expressing either WT or a mutant form of Sar1a (Figure 5C). Furthermore, a part of GFP-Sar1a WT colocalized with PLIN2 in OA-treated cells (Figure 5D), and neutral lipids in the cytosol were surrounded by GFP-Sar1a WT (Appendix A). The co-localization of GFP-Sar1a H79G and PLIN2, however, was not observed (Figure 5D), which is similar to Sar1b, suggesting the H79G mutant form of Sar1 may interfere with interaction of PLIN2 and LD. These results suggest that both Sar1a and Sar1b affect PLIN2 localization to LD in cells.

### 2.5. Reduction of PLIN2 Localization to LD in Sar1-Depleted Cells

To further investigate how endogenous Sar1 contributes to the localization of PLIN2 to LDs, the effect of Sar1 suppression by shRNA was examined. Because the two endogenous Sar1 proteins have highly similar amino acid sequences (97% homology), obtaining antibodies capable of distinguishing between the two is difficult. Thus, we studied the mRNA expression levels of *SAR1A* and *SAR1B* in cells using RT-PCR. *SAR1A* depletion by shRNA did not affect the expression levels of *SAR1B*, and vice versa (Figure 6A). Under the conditions, depletion of *SAR1A*, *SAR1B*, or both did not change intracellular PLIN2 protein levels, either with or without OA treatment (Figure 6B,C). Intracellular TG levels did not change upon the suppression of *SAR1A* or *SAR1B*, either with or without OA-treatment (Figure 6D). In cells treated with shRNA scramble (Scr), part of the PLIN2 signal showed ring-like structures in cytosolic space (Figure 6E). Such ring-like structure of PLIN2 was observed in the cells depleted of either *SAR1A* or *SAR1B*, while the fluorescent intensity of PLIN2 is slightly stronger than the others. In cells simultaneously depleted of both *SAR1A* and *SAR1B*, the intracellular PLIN2 signal scattered and did not form a circular ring-like structure (Figure 6E). Furthermore, the intracellular localization of PLIN2 in these cells was investigated using sucrose density gradient centrifugation. When both genes were depleted, the ratio of PLIN2 recovered in the LD fractions decreased significantly; however, neither *SAR1A* depletion nor *SAR1B* depletion changed the intracellular distribution of PLIN2 in cells (Figure 6F). These results suggested that both Sar1a and Sar1b are involved in PLIN2 localization to LDs in cells.

## 3. Discussion

In this study, we investigated the effects of Sar1a and Sar1b on numbers, size, and components of LDs in the human hepatoma cell line HuH7 cells. Sar1 is a well-known regulator of COPⅡ vesicle assembly on the ER membrane; however, little is known about its effects on LD formation and maturation. We hypothesized that Sar1, which acts as a regulator of vesicle formation, could also play a role in LD behavior. We isolated LD fractions using sucrose density gradient centrifugation and found that intracellular localization of PLIN2, but not PLIN3, to LD fractions decreased when Sar1b H79G was expressed (Figure 3B). The mislocalization of PLIN2 was verified by depleting endogenous *SAR1* mRNA (Figure 6). Abrogation of either *SAR1A* or *SAR1B* did not cause significant changes in PLIN2 localization to the LD, which corresponds well with the previous report describing how Sar1a and Sar1b could each compensate for a deficiency in the other [34]. These observations suggest that Sar1a and Sar1b play a regulatory role in PLIN2 localization to LDs.

Perilipins are major LD-associated proteins in cells and are believed to be crucial for LD formation [29]. PLIN2 has been shown to be a major LD surface protein in HuH7 hepatoma cells [16], and overexpression of PLIN2 causes LD accumulation in cells [5,6]. PLIN2-knockout mice show resistance to obesity and nonalcoholic steatohepatitis induced by a methionine-choline-deficient diet [35,36]. Overexpression of PLIN2 promotes lipid accumulation in fibroblasts and macrophages [37,38], as well as in HepG2 hepatic cells [39]. Recently, it was reported that PLIN2 overexpression protected LDs from autophagic degradation [40], and that LD-unbound PLIN2 is unstable and degraded via the ubiquitin-proteasome pathway [31,32]. Thus, understanding the intracellular distribution of PLIN2 is necessary to elucidate LD homeostasis.

Each one of PLIN2 and PLIN3, sharing sequence and functional similarities, can compensate for a deficiency in the other when acting as protective scaffold proteins for LDs [29]. In our results TG levels did not change while PLIN2 levels in LD fractions reduce, suggesting some proteins could compensate the decrease in PLIN2 on the LD surface to maintain the LD particles. The localization of PLIN3 differs from that of PLIN2, as PLIN3 was initially identified as a binding protein for mannose-6-phosphate receptor and is involved in the lysosome transport system. Our data showed that the distribution pattern of PLIN3 did change slightly in cells expressing Sar1a, but not in cells expressing Sar1b (Figure 3 and Figure 5A), indicating PLIN3 could compensate for the PLIN2reduction on LD surface. Recently, PLIN3 was shown to localize to LDs in RalA-GTPase-dependent manner [41]. Thus, there may be several GTPases that regulate the localization of LD components to LDs in cells. Several groups, including ours, have investigated LD protein profiles by proteomic analysis, and showed that actually wide variety of LD proteins was recovered in LD fraction [15,16,42,43]. It is reasonable to assume that some proteins could be recruited in addition to PLIN3 to LD when PLIN2 localization is reduced.

We observed the alteration of PLIN2 on the LD fraction by either expression of mutant form of Sar1 or Sar1 suppression; however, the effect of the Sar1b H79G mutant was somehow different from the other conditions. The total amount of intracellular PLIN2 protein levels decreased by approximately 30% in cells expressing Sar1b H79G, while cells expressing Sar1a H79G cells (Appendix A) and Sar1-depleted cells (Figure 6B,C) did not change the total PLIN2 levels. We considered a possibility that Sar1 might be involved in formation of mature LD particles through ER recruitment of LD components. The expression of Sar1 H79G mutant, but not the suppression of *SAR1B* mRNA, decreased PLIN2 in LD fraction, suggesting that the effect of Sar1 could be an indirect mechanism, namely, that Sar1b could interact with putative partner protein(s) to act on LD formation. We would like to investigate the partners of Sar1 in the cells in the future.

Sané et al. recently reported the possible roles of Sar1 in enterocytes using CRISPR-Cas9 technique to knockdown Sar1a and Sar1b in Caco-2 cells [44]. Interestingly, PLIN2 was decreased in *SAR1A* –/– cells, while *SAR1B* –/– upregulated PLIN2 expression. Since the chylomicron retention disease is caused by mutation of the *SAR1B* gene but not the *SAR1A* gene, and the patients show lipid accumulation in epithelial layer of intestine but not in liver, there must be an intestine-specific mechanism for Sar1 to mobilize PLIN2.

In contrast to the PLIN2 levels in the LD fraction, the TG levels in the LD fraction did not change significantly in cells expressing Sar1 H79G (Figure 3D and Figure 5C). Immunofluorescent assays using the LipidTox Red neutral lipid stain reagent showed that the number of LDs in cells expressing GFP-Sar1b WT tended to increase slightly; however, the ratio of total LD area to total cell area was not significantly different (Figure 1B and Appendix A). Taken together, these results indicate that Sar1 has some roles on PLIN2 localization to LDs during its maturation, but not in lipid accumulation in LD particles. PLIN2 is known to be replaced by PLIN1 during the maturation of adipocytes [45]. Thus, there might be a relationship between LD size and the type of PLIN proteins localized to LDs. The possible effect of Sar1b expression on LD size is interesting; however, further studies are needed to elucidate the regulation of LD size. Newly synthesized proteins and lipids in the ER are transported to the Golgi apparatus in a COPⅡ-coated vesicle trafficking pathway [18,19]. COPⅡ-coated vesicle assembly is initiated by the GDP-GTP exchange of Sar1 [20,21]. Intracellular transport vesicles containing very-low-density lipoproteins bud off from the ER in a Sar1b-dependent manner. However, other vesicles containing various proteins and lipids are released from the ER in a Sar1a-depndent manner [24]. Like transport vesicles, LDs form and then bud off from the ER membrane bilayer. Recent studies have elucidated that some critical proteins, including lipid droplet assembly factor-1 (LDAF-1), are involved in TG accumulation in the ER membrane bilayer and in nascent LD particles pinching off from it [11,46]. However, it has not been fully elucidated whether molecules related to vesicle budding also contribute to protein localization during LD formation. Some ER membrane proteins, such as UBXD8 and AAM-B, can also localize to LDs in a Sar1-independent manner [47]. In contrast, a previous study showed that ATGL localization to LD is COPⅡ-dependent [48]. Previously, we showed that Sec22b, a SNARE protein on the COPⅡ membrane, was found in the LD fractions [15]. In cells expressing Sar1 H79G mutant, Sec22b localization to LDs decreased compared to that in control and Sar1 WT expressing cells (Figure 3B and Figure 5A). This suggests that the COPⅡ vesicle is involved in PLIN2 localization to LD by the action of Sar1.

LD unbound PLIN2 is unstable and degraded via ubiquitin-proteasome pathway [31,32], and two types of E3 ligase (TEB, AIP4) have been shown to regulate PLIN2 degradation [49,50]. Degradation of PLIN2 proceeds when cellular LD is regressed under depletion of lipid sources. In this study, the number and size of LD did not change, since the cells were incubated in medium containing 10% FBS. PLIN2 in LD fraction decreases without a decrease in the total PLIN2 in Sar1a H79G expressed cells and in Sar1 suppressed cells. The distribution pattern of PLIN2 on the density gradient changed and parts of PLIN2 moved to intermediate fractions. It is possible that partial degradation of PLIN2 may occur in cells expressing Sar1b H79G, but further study is needed.

Like other GTPase, the ER localization of Sar1 is dependent on GTP binding [51,52]. Cytosolic Sar1, GDP binding form, is converted to the ER membrane-bound form by Sar1 GEF Sec12 [20]. GFP-tagged Sar1 WT proteins were found in the cytosol in untreated cells; however, in OA-treated cells, a part of Sar1 WT was found in a ring-like structure (Figure 1A and Appendix A). As expected, colocalization of PLIN2 and GFP-tagged Sar1 was observed in OA-treated cells (Figure 4 and Figure 5E). Considering that neither PLIN2 nor Sar1 contains transmembrane domains, it is possible that these proteins recognize certain scaffold proteins or lipids at the LD membrane for binding. A previous report showed the presence of Sar1 in the LD fraction by LC-MS/MS proteome analysis [17]. Although some proteome analyses of LD proteins have been reported, to the best of our knowledge, there are no Sar1-associated proteins in the list of LD-associated proteins. Thus, in near future, we would like to investigate the binding partners of PLIN2 and Sar1, either lipids or proteins, by examining the lipid composition of ER and LD membrane using LC-MS/MS.

In conclusion, we demonstrated that the PLIN2 LD localization pathway is regulated by Sar1 GTPase in cells. A future question is whether Sar1 regulates LD budding or LD maturation using COPⅡ vesicles. Further studies are necessary to elucidate the molecular mechanism underlying the localization of Sar1 GTPase and PLIN2 on the LD surface.

## 4. Materials and Methods

### 4.1. Cell Culture

HuH7 cells (purchased from ATCC) were grown in Dulbecco’s modified Eagle’s medium supplemented with 10% fetal bovine serum (Gibco, Waltham, MA, USA), 100 U/mL penicillin, and 100 μg/mL streptomycin at 37 °C in a 5% CO_2_ incubator. For LD formation, the cells were treated with 0.5 mM OA for 24 h at 37 °C in a 5% CO_2_ incubator.

### 4.2. Plasmid and Recombinant Adenovirus

cDNA fragments encoding open reading frames of human *SAR1A* and *SAR1B* were amplified from total RNA isolated from HuH7 cells using reverse transcription-PCR. The H79G mutation was introduced in the Sar1 proteins using Phusion polymerase (NEB, MA, USA) using primers (Sar1a H79G R primer: 5′ ttgctcTCCcccaccaagatcaaaagt 3′, Sar1b H79G F primer: 5′ ggtggaGGAgttcaagctcgaagagtg 3′, Sar1b H79G R primer: 5′ ttgaacTCCtccacccagatcaaaagt 3′). The inserts were sequenced, and the structures of all plasmids were confirmed by restriction analysis using ABI PRISM 3100 (Applied Biosystems, Waltham, MA, USA). Enhanced green fluorescence protein (eGFP) or Flag-tag was fused at the N-terminal of Sar1 proteins in all constructs used in this study. For RNA-interference (RNAi), short hairpin RNAs (shRNAs) containing the target sequences corresponding to *SAR1A* or *SAR1B* were introduced into the pSilencer 2.1-U6 Pro vector (Thermo, Waltham, MA, USA). The shRNA sequences are listed in Appendix A. All constructs were sub-cloned into the pENTR4 vector. For overexpression and RNAi, the cells were infected with each virus for 48 h and 72 h at 37 °C in a 5% CO_2_ incubator, respectively.

### 4.3. RT-PCR

Total RNA was isolated from the cells using Nucleospin RNA Ⅱ (Takara, Shiga, Japan). RNA samples were reverse-transcribed using MuLV reverse transcriptase (Applied Biosystems) in a total volume of 20 μL, and qRT-PCR was performed using the StepOnePlus Real-Time PCR systems (Applied Biosystems). Aliquots of the reverse transcription products were amplified in 20 μL of a reaction mixture containing PowerUp SYBR Green Master Mix (Applied Biosystems) and 0.5 μM each primer. The primer pairs used are listed in Appendix A.

### 4.4. Fluorescent Microscopy

Cells grown on coverslips were fixed in 4% paraformaldehyde (Nacalai tesque, Kyoto, Japan) for 10 min at room temperature, followed by being permeabilized with 0.1% (*w*/*v*) Triton X-100 in phosphate-buffered solution (PBS) for 10 min at room temperature. The permeabilized cells were blocked in PBS containing 1% bovine serum albumin at room temperature for 1 h. The cells were treated with anti-PLIN2 rabbit polyclonal antibody (Protein tech, Rosemont, IL, USA, dilution rate 1:500) in PBS at room temperature for 1 h. After washing with PBS, the cells were treated with Alexa Fluor 594-labeled goat anti-rabbit IgG (H + L) (Thermo, dilution rate 1:500) in PBS under the same condition. To visualize the LD and nuclei, the cells were stained with LipidTox Red (Invitrogen, Waltham, MA, USA) and DAPI (Dojindo, Kumamoto, Japan), respectively. The cells were observed under an FV10i confocal laser-scanning microscope (Olympus, Tokyo, Japan). We repeated the experiment three times and analyzed more than five cells per each experiment, for a total of at least 15 cells. The numbers and the area of LDs and the cytosol area were calculated using ImageJ software (NIH). The gray values shown in the line profile analysis were calculated using ImageJ software.

### 4.5. Subcellular Fraction

Subcellular fractionation was performed as described previously [15]. Briefly, HuH7 cells were suspended in buffer A (20 mM Tris (pH7.4) containing 1 mM EDTA, 250 mM sucrose, and protease inhibitor cocktail (Sigma-Aldrich, Saint Louis, MO, USA)) and then homogenized by 25 strokes using a 27 G needle. The samples were kept at 0–4 °C during the whole process. The homogenate was centrifuged at 3500× *g* for 10 min, and the supernatant was used as the post-nuclear fraction. The sucrose concentration of the supernatant was adjusted to 26%, then it was loaded in the middle of a 2–51% stepwise sucrose gradient (2%, 10%, 18%, the supernatant, 35%, 43%, and 51%) (Figure 3A). The gradient was centrifuged at 110,000× *g* for 2 h at 4 °C, using an RPS40T rotor (Hitachi Koki, Ibaraki, Japan), and aliquots were collected from the top of the gradient.

### 4.6. Immunoblotting

Cells were treated with lysis buffer (50 mM Tris (pH7.5), 150 mM NaCl, 0.1% (*w*/*v*) Triton X-100, and protease inhibitor cocktail) to prepare cell lysate. The protein concentration of cell lysates was determined using the BCA protein assay kit (Thermo). To prepare the cell lysate, cells were treated with lysis buffer and sonicated for 10 s on ice using UR-20P (Tomy Seiko, Tokyo, Japan), and centrifuged at 15,000 rpm at 4 ℃ for 10 min. We defined the supernatant as a cell lysate in this study. Cell lysates were subjected to SDS-PAGE, followed by immunoblotting. Bands were visualized using the ECL Prime Western Blotting System Cytiva (Cytiva, Waltham, MA, USA) and recorded by LAS500 (GE Healthcare, Waltham, MA, USA). The intensity of each immunoreactive band was measured by FIJI/ImageJ commercial software. Commercial antibodies used for immunoblotting and immunofluorescent microscopy were listed in Appendix A.

### 4.7. TG Measurement

Lipids were extracted from cells using the Folch method, followed by being re-suspended in isopropanol. The amount of TG of an aliquot (25%) was measured using the Triglyceride E test Wako (Wako, Osaka, Japan). TG levels were normalized based on the protein concentration of the same fraction.

### 4.8. Data Presentation and Statistical Analysis

Data are presented as a scattered dot plot or the mean ± SD. The results were analyzed using one-way analysis of variance with the Tukey-Kramer post hoc test using Easy R [53]. Statistical significance was set at *p* < 0.05.

## Figures and Tables

**Figure 1 ijms-23-06366-f001:**
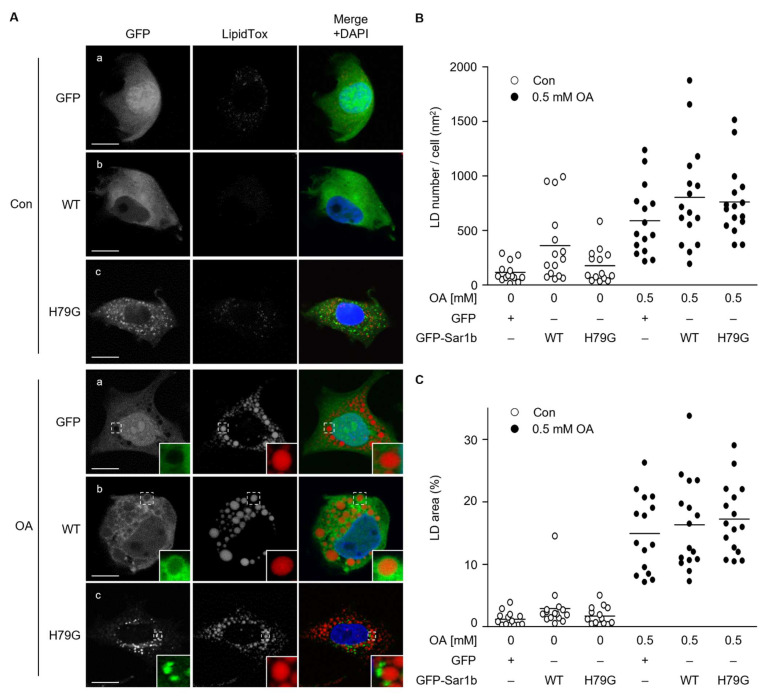
Peripheral localization of GFP-Sar1b WT on lipid droplet in oleic acid-treated cells. (**A**) HuH7 cells transiently expressing GFP (**a**) GFP-Sar1b WT (**b**) GFP-Sar1b H79G (**c**) were treated with (OA) or without (Con) 0.5 mM oleic acid for 24 h. The cells were stained with LipidTox and DAPI. Boxed areas are shown in higher magnification in the insets. Bar, 10 μm. Data are representative of experiments repeated at least three times. (**B**) The number of LDs per area in (**A**) are shown. Results are shown as a scattered dot plot. *p*-Values for GFP Con vs. GFP-Sar1b WT Con; and GFP OA vs. GFP-Sar1b WT OA were 0.350 and 0.417, respectively. (**C**) The ratio of LD area to total cell area in (**A**) are shown.

**Figure 2 ijms-23-06366-f002:**
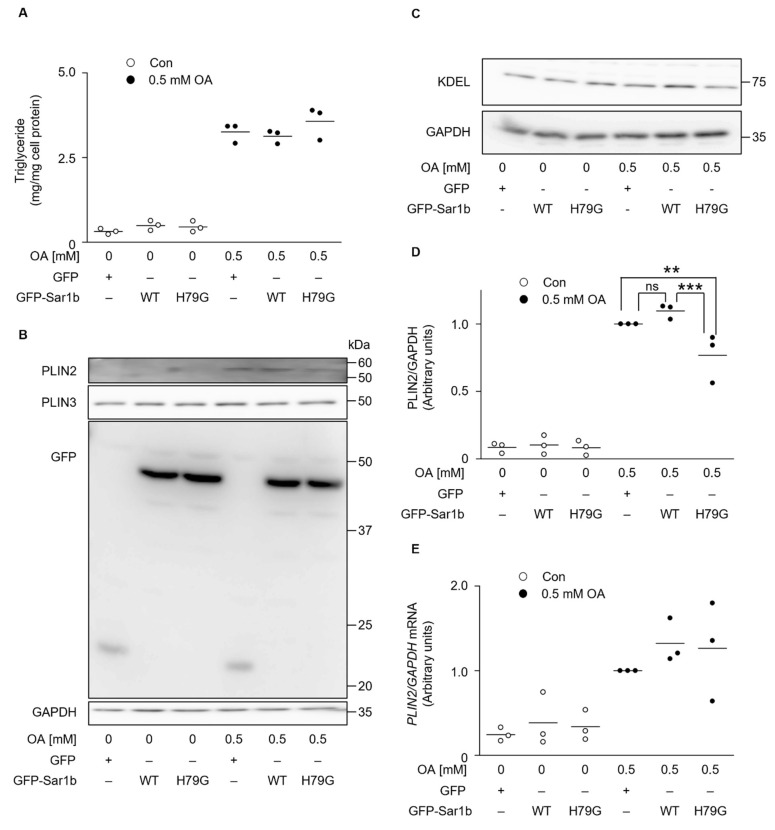
The intracellular PLIN2 protein levels decrease in cells expressing GFP-Sar1b H79G. HuH7 cells transiently expressing GFP, GFP-Sar1b WT, or GFP-Sar1b H79G were treated with (OA) or without (Con) 0.5 mM oleic acid for 24 h. (**A**) Intracellular triacylglycerol (TG) levels were quantified. Results are presented as a scattered dot plot. (**B**,**C**) Cell lysates were analyzed by western blotting using antibodies against the indicated proteins. Data are representative of experiments repeated at least three times. (**D**) The ratio of PLIN2 protein levels relative to GAPDH is expressed in arbitrary units. The PLIN2 levels in OA-treated cells expressing GFP were set to 1.0. The results are presented as a scattered dot plot. ** *p* < 0.01, *** *p* < 0.001 (vs. OA expressing GFP). (**E**) Total RNA was isolated from the cells. RT-PCR was performed to determine the mRNA levels of the indicated genes. The ratio of *PLIN2* mRNA levels relative to the *GAPDH* mRNA levels is shown as arbitrary units. *PLIN2* mRNA levels in OA-treated cells expressing GFP were set to 1.0. The results are presented as a scattered dot plot.

**Figure 3 ijms-23-06366-f003:**
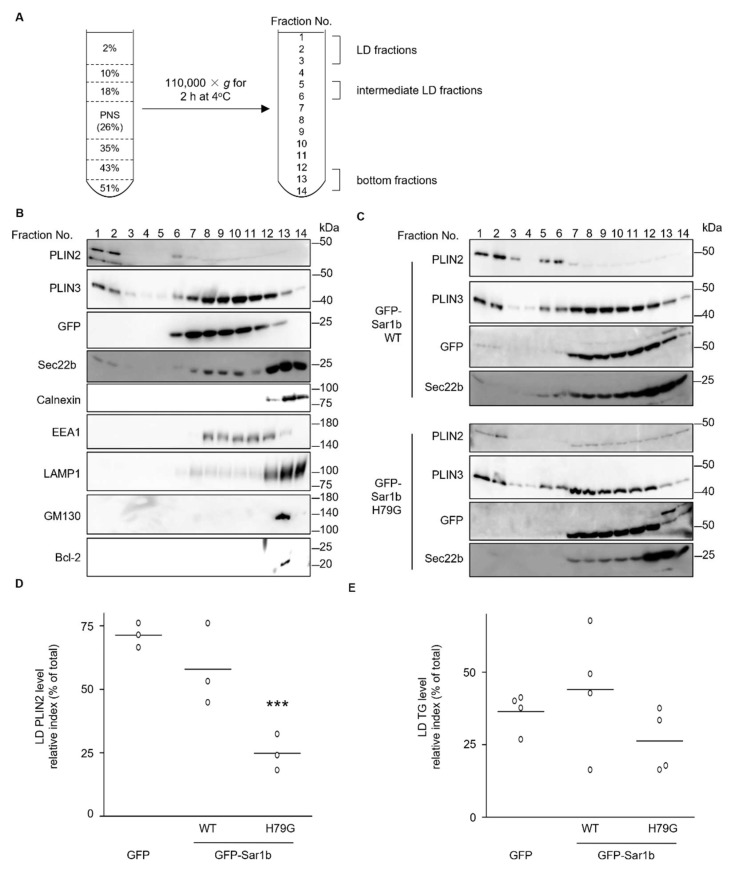
PLIN2 localization to lipid droplet in cells expressing WT or a mutant form of Sar1. (**A**) A scheme of sucrose density gradient centrifugation in this study. Post-nuclear supernatant (PNS) adjusted to 26% sucrose was placed in the middle of 2–51% gradient. After centrifugation, samples were recovered from the top into 14 fractions. Fractions 1 and 2, 5 and 6, and 12 to 14 were defined as LD fraction, intermediate LD fraction, and bottom fraction, respectively. The PNS fraction loaded before the centrifugation corresponds to fraction 7 to 10. (**B**,**C**) Lysates from cells treated with 0.5 mM oleic acid were fractionated by sucrose density gradient centrifugation, and an aliquot (10%) of each fraction was analyzed by western blotting using the antibodies to the indicated proteins. Lysates from cells expressing GFP (**B**), GFP-Sar1b WT, or GFP-Sar1b H79G (**C**) were fractionated. (**D**) The ratio of intensity of PLIN2 bands in the LD fractions (1 and 2) to that in sum of the all the lanes are shown as a scattered dot plot. *** *p* < 0.001 (vs GFP). (**E**) The ratio of TG levels in LD fractions to total cell lysate are shown as a scattered dot plot. *p*-Values for GFP vs. GFP-Sar1b WT; and GFP vs. GFP-Sar1b H79G were 0.919 and 0.487, respectively.

**Figure 4 ijms-23-06366-f004:**
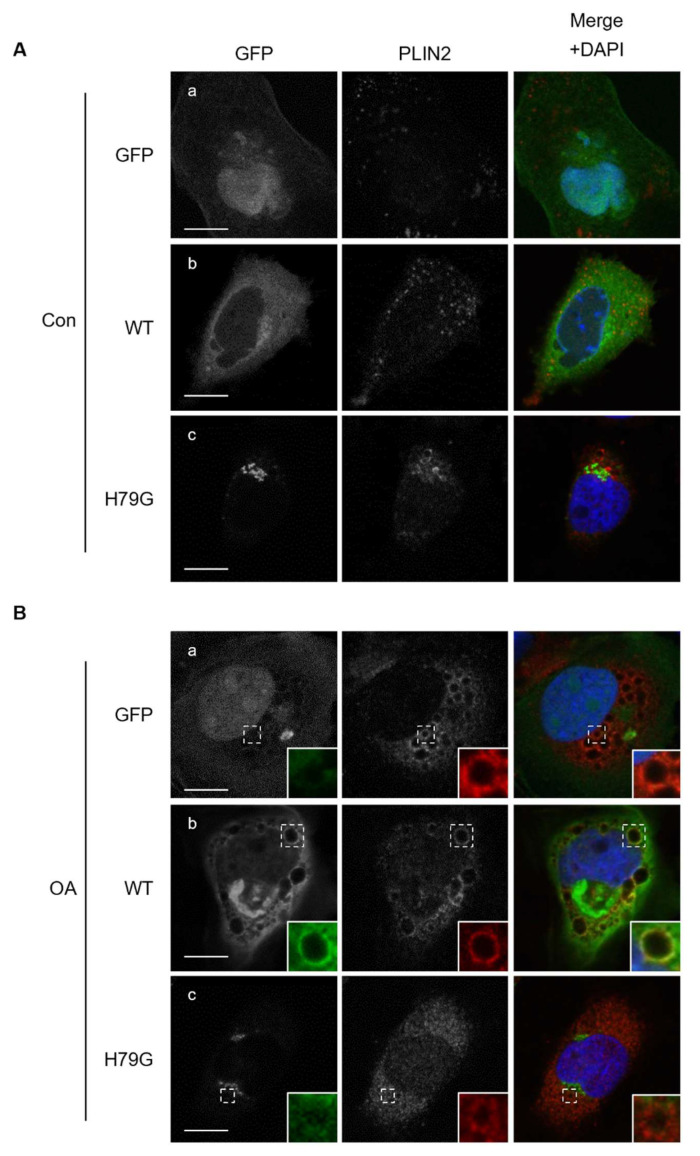
Partial colocalization between PLIN2 and GFP-Sar1b WT in lipid droplets. HuH7 cells transiently expressing GFP (**a**), GFP-Sar1b WT (**b**), GFP-Sar1b H79G (**c**) were treated with (**B**: OA) or without (**A**: Con) 0.5 mM oleic acid for 24 h. The cells were stained with the antibody against PLIN2 and DAPI. Boxed areas are shown in higher magnification in the insets. Bar, 10 μm. Data are representative of experiments repeated more than three times.

**Figure 5 ijms-23-06366-f005:**
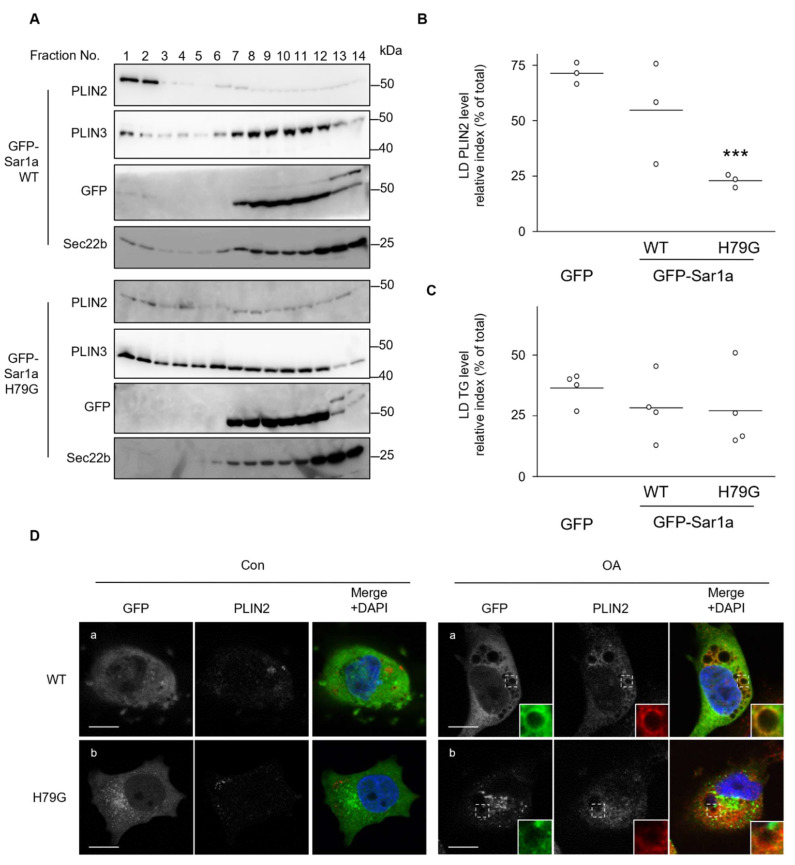
Effect of Sar1a on PLIN2 localization to lipid droplets. HuH7 cells transiently expressing GFP-Sar1a WT or GFP-Sar1a H79G were treated with 0.5 mM oleic acid (OA) for 24 h. (**A**) Lysates from cells containing LDs were fractionated by sucrose density gradient centrifugation, and an aliquot (10%) of each fraction was analyzed by western blotting using the antibodies against the indicated proteins. (**B**) The ratio of PLIN2 intensity in LD fractions to sum of all fractions are shown as a scattered dot plot. *** *p* < 0.001 (vs GFP). (**C**) The ratio of TG levels in LD fractions to total fractions is shown as a scattered dot plot. *p*-Values for GFP Con vs. GFP-Sar1a WT Con; and GFP vs. GFP-Sar1a were 0.518 and 0.546, respectively. (**D**) Cells expressing GFP-Sar1a WT (**a**) or GFP-Sar1a H79G (**b**) were stained with the antibody against PLIN2 and DAPI. Boxed areas are shown in higher magnification in the insets. Bar, 10 μm. Data are representative of experiments repeated at least three times.

**Figure 6 ijms-23-06366-f006:**
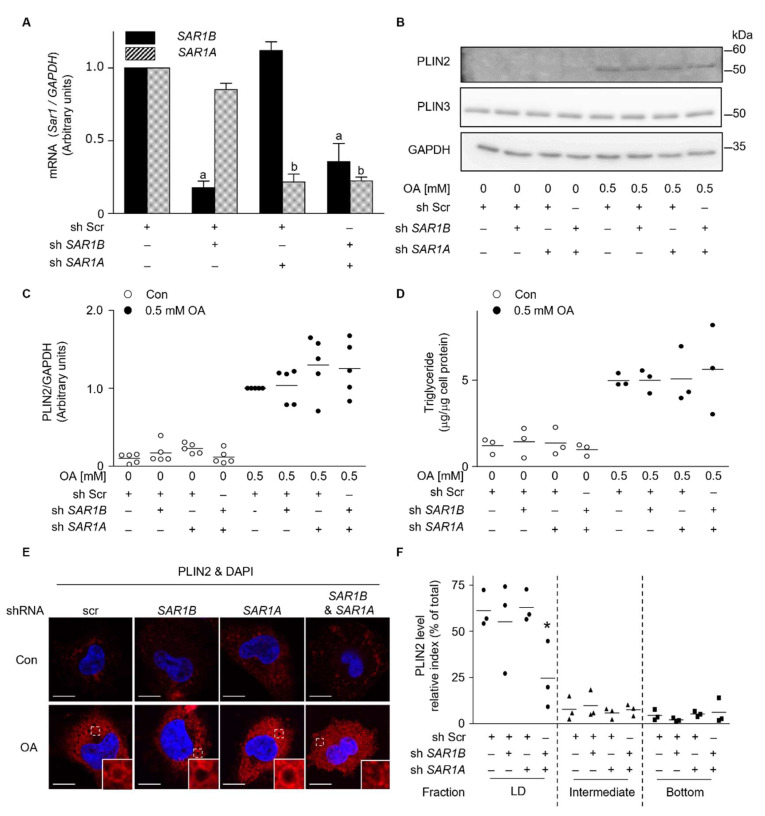
PLIN2 localization to lipid droplets attenuated in Sar1 depleted cells. HuH7 cells treated with shRNA scramble (Scr), sh *SAR1B*, sh *SAR1A*, or sh *SAR1B* and *SAR1A*, were incubated with (OA) or without (Con) 0.5 mM oleic acid for 24 h. (**A**) Total RNA was isolated from cells. RT-PCR was performed to determine the mRNA levels of the *SAR1A* and *SAR1B*. The ratio of *SAR1* mRNA level relative to the *GAPDH* mRNA level is shown in arbitrary units. *SAR1* mRNA levels in OA-treated cells were set to 1.0. The results are presented as the mean ± SD of three independent experiments. ^a^
*p* < 0.001 (vs. without sh *SAR1B*), ^b^
*p* < 0.01 (vs. without sh *SAR1A*) (**B**) PLIN2 and PLIN3 levels in cell lysates were analyzed by western blotting. Data are representative of experiments repeated at least three times. (**C**) The ratio of PLIN2 protein levels relative to GAPDH is expressed in arbitrary units. The PLIN2 levels in OA-treated cells expressing GFP was set to 1.0. The results are presented as a scattered dot plot. (**D**) Intracellular TG levels was quantified. Results are presented as a scattered dot plot. (**E**) The effect of depletion of *SAR1* mRNAs on PLIN2 distribution in cells was examined microscopically. Cells were stained with the antibody against PLIN2 and DAPI. Boxed areas are shown at higher magnification in the insets. Bar, 10 μm. Data are representative of experiments repeated at least three times. (**F**) Cell lysates from cells depleted *SAR1* mRNAs after treatment with 0.5 mM OA for 24 h were fractionated by sucrose density gradient centrifugation, and an aliquot (10%) of each fraction was analyzed by western blotting using the anti-PLIN2 antibody. Intensity of PLIN2 bands in the LD fraction (1 and 2), intermediate LD fraction (5 and 6), and bottom fraction (12, 13 and 14) to that in total fractions are shown as a scattered dot plot. * *p* < 0.05 (vs Scr).

## Data Availability

All data generated or analyzed during this study are included in this published article and its Appendix A.

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
