# Peer review of "Sar1 Affects the Localization of Perilipin 2 to Lipid Droplets"

_ijms, 2022, doi:10.3390/ijms23126366_

Round 1

Reviewer 1 Report

The manuscript entitled „Sar1 GTPase regulates the localization of perilipin 2 to lipid droplets“ by Makiyama et al., investigates the role of Sar1 in the recruitment of PLIN2 to lipid droplets (LDs).  LDs are intracellular organelles, consisting of a triacylglycerol (TG) core and a surrounding phospholipid monolayer. This monolayer is decorated with a variety of surface proteins such as members of the perilipin family, e.g. perilipin 2. Accumulation of TGs occurs in the phospholipid bilayer of the endoplasmatic reticulum (ER) and triggers the formation of nascent LDs that bud off from the ER. The transport mechanisms of LD proteins to their destination organelle remain largely unresolved.

In this manuscript, the authors are asking the main question if Sar1, an Arf family GTPase involved in COPII vesicle budding from the ER, could play a role in LD biogenesis. This hypothesis is based on the fundament that LDs as well as COPII vesicles are generated from the ER membrane. To address this question and to investigate the effect of the two Sar1 isoforms, Sar1a and Sar1b, on LDs they used the human hepatoma HuH7 cell line and either transiently expressed GFP-fusions of Sar1 isoforms or suppressed endogenous Sar1 by RNA-interference. Using fluorescence microscopy, westernblotting and subcellular fractionation they assessed the subcellular localization of Sar1 isoforms and PLIN2. They concluded that Sar1 and Plin2 colocalize on LDs upon OA treatment. Furthermore, they investigated the effects of the GTPase-deficient mutant version Sar1 H79G on PLIN2 localization and claim that Plin2 localization to LDs relies on the GTPase activity of Sar1. The main conclusion of the manuscript, as the title states, is that the enzymatic activity of Sar1 regulates the localization of Plin2 to LDs.

The manuscript is concisely written and structured. It addresses an important question that is potentially relevant to a broad readership from the fields of intracellular protein trafficking and organelle biogenesis. However, not all conclusion the authors made are fully supported by the provided data. Furthermore, the data/analysis quality itself is not satisfying in several experiments. The initial aim to investigate the role of Sar1 in LD formation was not satisfactorily addressed. The experiments are solely addressing the protein localization of either overexpressed forms of Sar1 or of Plin2. The results are therefore rather descriptive and correlative, and no molecular mechanism of how Sar1 could affect the subcellular localization of Plin2 or LD biogenesis in general was addressed. The title does not reflect the main findings of this manuscripts because the authors provide no evidence for a direct mechanistic interaction between Sar1 and Plin2. The authors need to add additional lines of experiments that address the underlying molecular mechanism. Alternatively, and at the very least, they need to significantly tone down their conclusions, including the title. A more descriptive title such as „Sar1 GTPase affects the localization of perilipin 2 to lipid droplets“ would be more appropriate.

Furthermore, the following issues need to be addressed before considering publication of this manuscript:

Major Issues:

1. The subcellular localization of GFP-tagged Sar1 WT isoforms and mutants, as well as of endogenous PLIN2 was assessed by fluorescent microscopy and upon biochemical fractionation. The authors show that WT forms of Sar1a and Sar1b colocalize with PLIN2 at LD upon OA of the cells. The provided data, however, raise a couple of questions:

- The localization of WT Sar1b under control conditions is inconsistent between different figures. In Figures 1A it appears cytosolic and in Figure 4 it is reminiscent of an ER pattern albeit the same experimental conditions apply to both figures. How is that possible?

- The localization of the Sar1b H79G mutant changes upon OA treatment to uncharacterized foci in Figure 1. In Figure 4, this mutant shows a similar loculation with and without OA treatment. Similarly, also for the Sar1a mutant different subcellular localizations are shown in different figures (Figures S3 panels c and Figure 5D). How is that possible? What are these foci and why does the Sar1 H79G mutant localization seem to be inconsistent?

- In Figure S3A, OA treatment panels a and b, it seems that the GFP-only and GFP-tagged WT Sar1a show a similar localization. Even the free GFP seems to be enriched around LDs and LD accumulation of WT Sar1a is not evident from the provided images.

2. The authors claim that the GTPase activity of Sar1 is required for correct PLIN2 localization to LDs. However, transient overexpression of GFP-tagged Sar1 H79G mutants leads to mislocalization in uncharacterized foci, which could cause several downstream problems in the cell that could indirectly affect the localization of PLIN2. The authors attempted to exclude “generic defects” in LD biogenesis by assessing LD number, LD area and the localization of another LD protein (PLIN3) and claim that they could rule out such effects. However, LD size and morphology clearly appear altered upon Sar1b H79G overexpression (Figure 1, OA condition, panel c compared to panels a and b). The quantifications in Figures 1B and 1C do not reflect what is shown in the representative image. Please clarify whether the image shown is not representative and revisit the quantification. Furthermore, it would be more appropriate to calculate LD size instead of the area they are occupying in cells. As from the image shown it appears that there are more and smaller LD compared to overexpression of WT Sar1b. In order to test whether recruitment of other LD proteins is also affected by Sar1 mutant expression, additional LD proteins such as ATGL and PLIN3 should be analyzed by fluorescence microscopy.

Regarding the localization of Plin3, the authors claim that the distribution pattern upon biochemical fractionation did not change in cells expressing Sar1a or Sar1b or the respective mutants. However, there is a difference in Plin3 distribution upon the expression of the Sar1a mutant (Fig 5 A). This an interesting observation, which the authors fail to describe in the results. They, however, speculate in the discussion that Plin3 could compensate the reduction of Plin2 on the LD surface. This is highly unlikely as PLIN2 appears diminished on LDs upon Sar1a and Sar1b mutant overexpression but Plin3 is only enriched on LDs upon Sar1a H79G but not upon Sar1b H79G overexpression. This needs to be discussed.

In order to strengthen the conclusion that the GTPase activity of Sar1 is indeed directly involved in the recruitment of Plin2 to LD, it would be necessary to test a mutant version that shows a similar subcellular localization as the WT form. Otherwise, simply the mislocalization of Sar1 could influence the localization of PLIN2. This could also be tested by redirecting WT Sar1 to another destination in the cell and by monitoring whether PLIN2 localization is altered as well.

3. The authors claim that WT Sar1a and Sar1b are required for the correct LD localization of PLIN2. However, the experiments addressing the expression and localization of Plin2 are often lacking sufficient data quality. They claim that the Plin2 fluorescent signal is “blurred” or “dot-like” and altered on LDs when transiently overexpressing the Sar1 mutants or depleting Sar1 in cells (Fig 4 OA, panel c ; Fig 5 D, OA panel b; Fig 6E Sar1a & Sar1b). It would be necessary to include a Lipidtox staining into these panels in order to assess whether the areas of interest are really representing LDs. Furthermore, it appears that the overall signal intensity of PLIN2 is reduced in the IF images when the Sar1 mutants were expressed (Fig. 4 and 5) and when Sar1 isoforms were depleted from cells (Fig. 6), which is inconsistent with the Western Blot data they provide. Only for Sar1b mutant expression they claim that endogenous PLIN2 amounts are reduced (Fig 2B). For the Sar1a mutant expression and for the depletion experiments, similar PLIN2 levels were detected by WB. How reliable are these data and why are the Western Blot data inconsistent with the IF images?

Importantly, for Fig 2B, the provided full scans of the blots in the supplement do not seem to be identical with the blots provided in the main figures. Have the blots been stretched? GAPDH was used as a loading control and for normalization of PLIN2 signals but from the full scans in the supplement it appears that the GAPDH signals and PLIN2 signals are not derived from the same membrane. If true, this would preclude such a normalization and raises the question whether Sar1b H97G expression really diminishes PLIN2 levels in cells. Please explain and revisit the original data and quantifications.

The biochemical fractionation of cells is supposed to provide the strongest evidence that less PLIN2 accumulates in LD fractions when Sar1 mutants were expressed (Figures 3 and 5). While there is indeed less PLIN2 in these fractions, there seems to be overall less PLIN2 in all fractions, which is, at least for Sar1a H97G, again inconsistent with the total protein blot in Fig. S4 and which also raises the question whether PLIN2 is specifically depleted from LDs or just reduced in total abundance in cells. Quantifications in Figures 3C and 5B are supposed to reflect selective reduction of PLIN2 on LDs but it is unclear how these calculations were performed. A similar analysis was performed upon Sar1 isoform depletion in Figure 6F but it seems that the sum of signals per sample (for example the black bars for the double-knock down sample) does not match 100%. If the overall PLIN2 amount is unchanged upon OA treatment under all conditions as shown in Fig 6B, the sum of signals from the individual fractions should be 100% in all cases. The original blot data are missing for these experiments and should be included into the supplement.

4. For their statistical analyses, the authors usually test significance between the Sar1 mutant samples and the GFP-only control. It would, however, be more appropriate to compare WT and mutant samples. Please recalculate these statistics and indicate in the figure to which sample comparisons the significance asterisks belong.

5. As outlined above, the data provided in this manuscript only address a correlation between Sar1 expression and PLIN2 localization to LDs. No mechanistic insight is provided or even addressed. It is surprising that the overexpression of WT Sar1 isoforms does not seem to show any positive effect of PLIN2 localization in cells, which would argue that endogenous Sar1 levels are sufficient. In contrast, overexpression of the Sar1 mutants seem to have a dominant-negative effect on PLIN2 localization. Is endogenous Sar1 suppressed selectively when the mutant versions are overexpressed? If not, one could assume that endogenous WT Sar1 was sufficient for correct PLIN2 localization, which would in turn raises the question how direct and specific the observed effects with the mutants are and whether the GTPase function of Sar1 was indeed required. To exclude overexpression artefacts, the authors should determine the degree of overexpression and ensure that Sar1 WT and mutants are expressed to a similar extend.

The knock-down experiments of endogenous Sar1 isoforms address the necessity of endogenous Sar1 but as outlined above the data quality in Figure 6 is not yet sufficient to assess whether PLIN2 was indeed selectively depleted from LDs.

6. Regarding the Sar1-knock down experiments in Figure 6, the authors confirmed the depletion of Sar1 isoforms only by qRT-PCR on transcript level. This is not necessarily representing the protein level in the cells and Western blotting analyses should be included to assess the endogenous Sar1 protein level. It would be interesting to repeat the experiment with Sar1 knockout cells. In a previous publication (PMID: 31409740), Sar1 double knockout cells were characterized and it appears that PLIN2 was significantly upregulated in these cells. This would be inconsistent with the double-knockdown experiments in this manuscript and a side-by-side comparison of knock-out and knock-down cells could significantly strengthen this manuscript. At least, the authors need to discuss the previously published data on Sar1 and PLIN2 in context of their findings.

Minor Issues:

1. Regarding data presentation and quantification, a more transparent visualization would be helpful. A box plot replacing the bar graphs (e.g. Fig 1 B + C, Fig 2 A,C + D) in the figures would allow assessment of the variation between the single experimental repeats.

2. It would be helpful, if the authors could include a rational why they focused on the investigation of Plin2 as a LD protein in this manuscript. It is not intuitive for the reader to follow this.

3. The Sar1 mutant H79G should be introduced and discussed better. What is known about its activity and its subcellular localization?

4. The microscopy-based experiments include a GFP-only control. To exclude that GFP expression itself interferes with LD biogenesis, number and size, the authors should include non-transfected cells as negative controls.

5. The method section is incomplete. Which transfection method was used to transiently express the constructs in the cell? How was the OA treatment performed? How exactly was the TG measurement performed?

6. It would be helpful to add a schematic/cartoon depicting the biochemical fractionation paradigm and indicating which fraction numbers correspond to which density in the gradient.

7. There is one generic flaw in the manuscript. The authors write “Plin2 relative expression levels in the LD fraction”. This needs to be replaced by “Plin2 localization/amount in the LD fraction” (e. g. Lines 135, 179, 227). Plin2 is not expressed in LD fractions, it is targeted there.

8. Lane 146 font size

9. Lane 160 Misslabeling of figure: FigS3A

10. Lane 276 UBXD8 and AAM-B are no transmembrane proteins

11. Lane 320 nucleospin

12. Figure 4 : Labeling A and B are missing

Reviewer 2 Report

Summary:

In this paper Makiyama et al. investigate how Sar1 GTPase, a component of COPII-mediated vesicle trafficking, affects lipid droplets in hepatoma cells. They found that Sar1 is required for proper lipid droplet localization of PLIN2. Overall, the hypothesis of this paper is clear and the experimental approaches are straightforward. Yet, a few technical improvements and alternative analyses would make the study more comprehensive.

Major point:

Effect of Sar1 on LD number, area, and size. The authors claim that expression of WT Sar1b or Sar1b H79G does not have statistically significant effects on lipid droplet number and area. This is based on data shown in Figures 1B-C, which show a statistically insignificant increase in LD number/area upon expression of WT Sar1b. I recommend to increase the number of cells analyzed to clarify if this effect indeed occurred by chance. At least in the picture presented in Figure 1A it appears that cells expressing the Sar1b H79G have smaller lipid droplets compared to cells expressing GFP or WT Sar1b. To clarify these issues, I suggest to increase the number of cells analyzed and to quantify lipid droplet size in addition to number and area.

Overexpression of Sar1b H79G reduces total PLIN2 levels, which is apparently due to the loss of LD-associated PLIN2. While knockdown of Sar1a + Sar1b leads to a similar reduction in LD-associated PLIN2 it does not affect total cellular PLIN2. Why? According to the subcellular fractionation experiments a substantial fraction of PLIN2 resides in the LD fraction and thus it seems intuitive that affecting LD-associated PLIN2 would affect total PLIN2. There is also a discrepancy between WB and IHC as it seems that total cellular PLIN2 signal is much lower after knockdown of Sar1a+Sar1b when detected by IHC but not by WB. These discrepancies should be clarified.

Overall, it is quite surprising that the authors do not detect changes in LD morphology/abundance or alterations in triglyceride content upon manipulation of Sar1 activity/expression. I would assume that loss of lipid droplet PLIN2 would increase triglyceride breakdown. It would be great to have this point clearly addressed in the discussion.

In an early paper the authors demonstrated that PLIN2 is degraded via the proteasome (Masuda et al. JLR 2006). I wonder if Sar1 affects LD localization of PLIN2 indirectly by altering PLIN2 degradation. It is certainly easy to test this.  

Minor points:

Image quality: LDs are often not visible due to inferior contrast of the red channel. One possibility to improve this is to use grayscale images whenever the single channels are presented and colored pictures only in the merged images. Insets often do not entirely match the boxed area. This should be corrected.

What is the rationale behind alternatively showing SEM and SD? In my opinion, the use of SD would illustrate the deviation of the datapoints better.

Reviewer 3 Report

In the reviewed paper, the researchers studied the effects of Sar1a and Sar1b on numbers, size and components of LDs in HuH7 cell line. The presented results allowed to indicate that PLIN2 LD localization pathway is regulated by Sar1. However, there were some shortcomings in the work. Authors did not explain sufficiently why they decided to use such a research model and why the introduced H79G mutation is crucial. It seems reasonable to use additional cell model in order to compare and finally confirm the conclusions. The figures descriptions are not very clear and require significant redrafting. Fig1, panel A, in the last microphotography representing H79G, the area that has been enlarged has not been marked. The authors do not sufficiently indicated statistically significant differences and it is not clear to me why SD and SEM were used interchangeably. It requires clarification and systematization throughout the paper. Fig.2 shows the ration of PLIN2 mRNA to the housekeeping GAPDH as reference gene, while it is nowhere shown whether an additional reference gene was used. Currently, it is required to use at least 2 reference genes in this type of study. The blots presented in Fig.3 are of poor quality and hardly legible, which would require correction. The references need to be unified because on page 5 there is a citation in the author data format and not in the numerical as in the entire text. The reviewed work is very interesting, but it loses due to many ambiguities that should be clarified.  

Round 2

Reviewer 1 Report

The authors addressed most of the concerns and the revised version of the manuscript is much improved. In Figures 4Ab and 4Bc, however, the images showing single channels do not match the merged images or the insets, respectively. This needs to be fixed. Upon correction, the manuscript may be accepted.

Author Response

Reviewer1

Thank you for your reviewing our manuscript again.

To improve our manuscript much more, we revised some points where the reviewer 1 pointed out. As you pointed out, the merged image in Fig 4Ab was different from that of a single channel. We apologize for such a mistake, and we have replaced it with the correct one. In Fig4Bc, we changed the size of the boxed area to match the insets.

Reviewer 2 Report

I appreciate the efforts of the authors to improve data presentation. In particular, scattered dot blots are helpful to estimate variation between samples.

Image quality: I encourage the authors to increase the brightness at least of the grayscale channels because the structures are hard to see.

In some figures, dot sizes are different between groups (e.g. Fig 6c).

Author Response

Reviewer2

Thank you for your reviewing our manuscript again.

To improve our manuscript further, we changed the contrast of the figure images to make the readers easily find out the difference between GFP, GFP-Sar1, and GFP-Sar1 H79G (increase the contrast with the same quantity in all grayscale images). 

In fig6.C and Fig.6D, some dots sizes were different in the same graph, thus we changed the dots size to make them equal in the same graph.